# Development of a Rapid LC-MS/MS Method for Simultaneous Quantification of Donepezil and Tadalafil in Rat Plasma: Its Application in a Pharmacokinetic Interaction Study after Oral Administration in Rats

**DOI:** 10.3390/molecules28052352

**Published:** 2023-03-03

**Authors:** Jiyoung Yoon, Doowon Choi, Wang-Seob Shim, Sanghee Choi, Yeo Jin Choi, Soo-Heui Paik, Kyung-Tae Lee

**Affiliations:** 1Department of Pharmaceutical Biochemistry, College of Pharmacy, Kyung Hee University, Seoul 02447, Republic of Korea; 2Department of Biomedical and Pharmaceutical Science, Graduate School, Kyung Hee University, Seoul 02447, Republic of Korea; 3Kung Hee Drug Analysis Center, College of Pharmacy, Kyung Hee University, Seoul 02447, Republic of Korea; 4College of Pharmacy, Suncheon National University, Suncheon 57922, Republic of Korea

**Keywords:** donepezil, tadalafil, lansoprazole, LC-MS/MS, validation, protein precipitation, pharmacokinetics, rat

## Abstract

This study aimed to establish a simple and sensitive analytical method to simultaneously quantify donepezil (DPZ) and tadalafil (TAD) in rat plasma using lansoprazole (LPZ) as an internal standard (IS) by using liquid chromatography tandem mass spectrometry. The fragmentation pattern of DPZ, TAD, and IS was elucidated using multiple reaction monitoring in electrospray ionization positive ion mode for the quantification of precursor to production at *m*/*z* 380.1 → 91.2 for DPZ, *m*/*z* 390.2 → 268.1 for TAD, and *m*/*z* 370.3 → 252.0 for LPZ. The extracted DPZ and TAD from plasma using acetonitrile-induced protein precipitation was separated using Kinetex C18 (100 × 2.1 mm, 2.6 µm) column with a gradient mobile phase system consisting of 2 mM ammonium acetate and 0.1% formic acid in acetonitrile at a flow rate of 0.25 mL/min for 4 min. The selectivity, lower limit of quantification, linearity, precision, accuracy, stability, recovery, and matrix effect of this developed method was validated according to the guidelines of the U.S. Food and Drug Administration and the Ministry of Food and Drug Safety of Korea. The established method achieved acceptance criteria in all validation parameters, ensuring reliability, reproducibility, and accuracy, and was successfully implemented in a pharmacokinetic study on the co-administration of DPZ and TAD orally in rats.

## 1. Introduction

Alzheimer’s disease (AD) is a common geriatric illness characterized by neurodegeneration associated with a reduction in cholinergic system activity in the cerebral cortex and other areas of the brain [1]. The cholinergic system plays crucial roles in physiological processes, such as attention, memory, sensory information, learning, stress response, wakefulness, and sleep. Acetylcholine (ACh) is a chemical messenger, neurotransmitter, and neuromodulator that participates in cholinergic system signaling and regulation. As cholinergic and adrenergic signaling is dysregulated owing to substantially reduced ACh synthesis in patients with AD, cholinesterase (ChE) inhibitors are prescribed as the first-line agents for AD management to increase ACh concentration by inhibiting hydrolyzing enzymes, such as acetylcholinesterase (AChE) and butyrylcholinesterase. Donepezil (donepezil hydrochloride, E-2020, arycept™, Eisai, DPZ) is the first selective AChE inhibitor approved for AD management [2].

As the blood–brain barrier (BBB) restricts drug delivery and absorption into the brain, the most critical determinant of successful DPZ is the enhancement of drug delivery into the brain [3,4]. However, recent studies have shown that BBB breakdown and cerebral hypoperfusion are commonly observed in patients with AD, and these pathophysiological changes not only induce neurodegeneration and cognitive impairment but also impair drug delivery into the brain [5]. Thus, the development of a delivery strategy that considers pathophysiological changes is imperative to optimize DPZ. A recent approach to improve pharmacotherapy in patients with AD is the concomitant administration of a low-dose phosphodiesterase type 5 (PDE5) inhibitor with DPZ [6]. Low-dose tadalafil (tadalafil, IC351, Cialis™, Lilly ICOS LLC, TAD), a selective PDE5 inhibitor with the longest duration of action, not only induces vasodilation by increasing cGMP and intracellular nitric oxide levels but also improves cognitive functions, thereby reducing memory impairment and anxiety in patients [7,8,9,10,11,12,13,14,15,16,17,18,19]. Moreover, increased cerebral blood flow was reported with TAD administration in the elderly with small vessel disease [20], which may imply that concurrent administration of TAD with DPZ may increase DPZ delivery and absorption by improving brain perfusion. Hence, combination therapy with DPZ and low-dose TAD may have synergistic pharmacodynamic effects to improve cognitive function in patients with AD.

The major concern with combination therapy with DPZ and low-dose TAD is the potential drug–drug interaction (DDI), as both drugs are metabolized by cytochrome p450 (CYP) 3A4 [21,22]. However, studies evaluating DDI between DPZ and TAD are limited. Methods for estimating DNZ or TAD using high-pressure liquid chromatography (HPLC) with UV [23,24,25] or fluorescence detection [26] have been previously reported. However, these methods have some limitations, given the low sensitivity and complicated sample processing procedure. In addition, previous studies have described liquid chromatography with tandem mass spectrometry (LC-MS/MS) detection, which was reportedly simpler and more sensitive for analyzing DNZ or TAD in rat plasma [27,28,29]. Hence, the establishment of a novel simultaneous analytical method to quantify DPZ and TAD in plasma samples is crucial for high-throughput and reliable sample analysis to conduct large-scale pharmacokinetic studies and routinely measure plasma DNZ and TAD concentrations in the DDI from pharmacokinetic studies, and only a simultaneous quantification method for DPZ is currently available [30,31,32]. To our knowledge, no sensitive and reproducible LC-MS/MS method for simultaneous analysis has been developed for determining DNP and TAD in rat plasma samples. Therefore, this study aimed to establish and validate simultaneous quantification methods for both DPZ and TAD in plasma samples using LC-MS/MS to investigate the potential DDI between DPZ and TAD.

## 2. Results and Discussion

### 2.1. Method Development

#### 2.1.1. Mass Spectrometric Analysis

The mass spectrometer parameters were optimized by infusing a solution of DPZ, TAD, and LPZ (IS) dissolved in 100% methanol. The infusion was directly performed in the ionization source using a syringe pump operating at a flow rate of 10 μL/min. The maximum amount of production was attained in the positive mode using a turbo ion spray electrospray ionization (ESI) interface. As a result of the full scan of Q1, DPZ was performed at a resolution of *m*/*z* 380.1 → 91.2, TAD at *m*/*z* 390.2 → 268.1, and LPZ at *m*/*z* 370.3 → 252.0 production (Figure 1).

#### 2.1.2. Chromatographic Conditions

The chromatographic conditions were applied to induce ideal separation and resolution among DPZ, TAD, and LPZ (IS). Various reversed-phase columns, mobile phase compositions of isocratic and gradient shapes, and flow rates were tested to optimize the separation conditions for the target analytes and IS. In our present study, Phenomenex Luna C18 (50 × 2.0 mm, 3 µm), Phenomenex Kinetex C18 (100 × 2.1 mm, 2.6 µm), and Halo C18 (50 × 2.1 mm, 2.7 µm) columns were tested to determine optimal separation conditions. Among the various columns tested for the analysis, Phenomenex Kinetex C18 (100 × 2.1 mm, 2.6 µm) provided the most appropriate separation of the matrix, high sensitivity, peak shape, and stable reproducibility, whereas other columns had low sensitivity, poor separation, peak tailing, or fronting. Isocratic elution was first utilized to separate the analytes; however, it was difficult to separate the peaks of the analyte and IS owing to the lack of power of the isocratic mobile phase. Therefore, gradient elution was selected to achieve optimal separation and good resolution. As LPZ has been reported to be unstable at low pH [33,34], we designed solvent A with 2 mM ammonium acetate to improve the stability of LPZ and solvent B with 0.1% formic acid in acetonitrile to optimize the peak shape. Acetonitrile was selected as a modifier because it exhibits a higher dissolution capacity from the matrix peak and efficient peak intensity compared to other organic solvents.

#### 2.1.3. Plasma Sample Preparation

DPZ is mostly extracted by using the liquid–liquid extraction (LLE) method using methyl tert-butyl ether, ethyl acetate, and n-hexane as the extraction solvents [27,35,36,37]. However, TAD is usually extracted using the solid phase extraction and LLE methods using dimethyl ether and dichloromethane as the extraction solvents [17,18,19,20,21]. For simultaneous extraction of DPZ, TAD, and IS with different chemical structural properties, protein precipitation was selected as an extraction method because it is a simple sample preparation procedure with excellent reproducibility [38,39,40]. Acetonitrile and methanol were tested as the precipitants, and a 6.7:1 ratio of acetonitrile and plasma consistently yielded high recoveries with symmetric peak shapes for DPZ, TAD, and IS without matrix interference when compared to those achieved with methanol. Thus, acetonitrile was selected as the precipitant for this experiment.

### 2.2. Analytical Method Validation

#### 2.2.1. Specificity and Lower Limit of Quantitation

To verify the selectivity, we utilized six different blank plasmas and pooled the blank plasma at each step. Figure 2 represents the chromatograms of (A) blank rat plasma; (B) LPZ (IS) (20 ng/mL) spiked blank plasma; (C) DPZ (200 ng/mL) spiked blank plasma; (D) TAD (200 ng/mL) spiked blank plasma; (E,F) blank plasma spiked with DPZ (1 ng/mL), TAD (1 ng/mL), and IS (20 ng/mL); and (G) plasma sample from 1 h after oral administration of DPZ (3 mg/kg) and TAD (0.3 mg/kg) with additional IS. During the retention time, the matrix peaks of DPZ (t = 1.32 min), TAD (t = 2.57 min), and IS (t = 2.37 min) did not interfere with each other. The LLOQ of DPZ (1 ng/mL) and TAD (1 ng/mL) were determined with signal-to-noise (S/N) ratios of >10, suggesting a sufficiently sensitive method to quantitate DPZ and TAD in plasma after the oral administration of DPZ and TAD to rats. The precision and accuracy of the LLOQ samples were ensured for the PK study.

#### 2.2.2. Linearity

Using linear regression, calibration curves for DPZ and TAD were plotted with seven concentrations over the range of 1–200 ng/mL. The equation for the calibration curves (*n* = 5) with the mean ± SD of the slope and the intercept was *y =* 0.277 (± 0.027) *x* + 0.033 (±0.007) (r^2^ ≥ 0.9904) and *y* = 0.0402 (± 0.0027) *x* + 0.001 (±0.003) (r^2^ ≥ 0.9966) for DNZ and TAD, respectively. The average slope and intercept of the regression equations were 0.2770 (co-efficient of variation (CV): 8.6%) for DPZ and 0.0403 (CV: 6.0%) for TAD. The linearity was satisfactory and reproducible over time.

#### 2.2.3. Precision and Accuracy

The precision and accuracy of the intra- and inter-day analyses are summarized in Table 1. For DPZ, the intra-day precision ranged from 1.06% to 2.26%, whereas the accuracy ranged from 86.93% to 102.17%. The inter-day precision and accuracy ranged from 3.14% to 4.75% and 89.08% to 102.53%, respectively. TAD, intra-day precision, and accuracy ranged from 1.90% to 5.52% and 95.80% to 99.66%, respectively. The inter-day precision and accuracy ranged from 3.42% to 8.19% and 96.40% to 99.48%, respectively. These results confirm the accuracy and precision of the simulated quantification method over a wide concentration range of the assay and satisfied the precision and accuracy ranges (%), as specified in the guidance of the MFDS and the FDA for bioanalytical applications [41,42].

#### 2.2.4. Stability

The stability of the stock solutions and rat plasma samples of DPZ and TAD under various conditions is shown in Table 2. The peak area response of the stored samples was directly compared with that of fresh stock solutions. The stability of the stock solution at room temperature (3 h) was 103.43% for DPZ and 107.75% for TAD. In addition, the stock solution stability under refrigeration (4 °C) for 14 days was 103.53% for DPZ and 106.30% for TAD. The plasma stability of DPZ and TAD was assessed by analyzing three replicate samples at two QC levels (2 and 160 ng/mL) under the following storage conditions. The stability of DPZ and TAD in rat plasma was maintained without apparent loss under various storage and handling conditions: room temperature (5 h), refrigeration (4 °C) for 14 days, four freeze/thaw cycles, extraction in an autosampler (10 °C, 20 h), and frozen conditions (−70 °C) for 4 weeks, which implies the long-term stability of DPZ and TAD. Therefore, based on the deviations from the nominal concentration within ±15%, DPZ and TAD were considered stable in rat plasma under all examined conditions without substantial degradation.

#### 2.2.5. Recovery and Matrix Effect

Table 3 highlights the recoveries of DPZ and TAD at three QC concentration levels (2, 16, and 160 ng/mL, *n* = 6), along with IS recovery (20 ng/mL, *n* = 6). The extraction recovery was evaluated by comparing the mean area responses of the two analytes spiked before extraction with those spiked after extraction. The matrix effect was evaluated by comparing the mean area responses of the two analytes spiked after extraction with neat solution samples at two QC levels (2 and 160 ng/mL, *n* = 6) for DPZ and TAD. The absolute extraction recoveries for DPZ and TAD were 90.86–95.99% and 96.17–98.59%, respectively. The extraction recovery of IS was 97.96%. The mean extraction recovery for DNA and TAD in rat plasma was greater than 90% at all three tested quality control (QC) concentrations with CVs of 15% or less (Table 3), suggesting high reproducibility and excellent analyte recovery using our one-step sample preparation method with PP.

According to previous studies, the effects of the matrix on extraction recovery, ion enhancement, and ion suppression should be critically assessed to develop a valid bioanalytical method using LC-MS/MS [15,16]. Therefore, the matrix effects for DPZ and TAD were 97.05–108.78% and 56.16–58.46%, respectively. The matrix effect of IS was 90.90%, and no significant differences in peak areas were detected. Even the mean matrix effects for TAD in rat plasma are less than 60%, with the corresponding CV (%) values of less than 15% at the two QC concentrations, suggesting acceptable and reproducible matrix effects between the analyte and endogenous substances in rat plasma.

### 2.3. Analytical Method Implementation to DDI PK Study in Rats

The established LC-MS/MS analytical method was successfully implemented to simultaneously quantify the PK parameters of DPZ and TAD in plasma after oral administration of DPZ and TAD, either alone or in combination in rats. The mean plasma concentration–time curves of DPZ and TAD are depicted in Figure 3. The PK parameters of DPZ and TAD are summarized in Table 4. The *C*_max_ of single DPZ administration was 59.776 ng/mL, achieved at a very short *T*_max_ of 0.5 h with a high AUC_last_ of 129.583 h·ng/mL. *C*_max_ of DPZ in combination treatment with TAD was 39.235 ng/mL, and the AUC_last_ was 100.5 h·ng/mL, which was also attained in a very short time (*T*_max_ = 0.5 h). The *C*_max_ of a single TAD administration was 17.795 ng/mL, and the *C*_max_ of TAD in combination with DPZ was 13.750 ng/mL. In addition, the AUC_last_ of a single TAD administration was 46.987 h·ng/mL, and the AUC_last_ of the combination was 34.419 h·ng/mL. Both groups exhibited very short *T*_max_ values (0.5 h). Our test of the longer *T*_1/2_ and *MRT*_last_ (mean ± SD; 2.630 ± 1.302 h and 2.617 ± 0.923 for single DAZ, respectively, vs. 3.793 ± 2.032 h and 3.735 ± 1.602 h for DPZ in combination with TAD, respectively) were observed. In addition, higher CL/F was found as 7.355 ± 4.516 L/h and 11.819 ± 9.852 L/h for single TAD and for TAD in combination with DPZ, respectively. A comparison of the plasma levels of DPZ and TAD between single and combination administration showed that a single administration of each drug achieved higher values than those of the combination groups. However, the differences in the mean plasma concentration and PK parameters between the single and combination groups of DPZ and TAD were statistically insignificant (*p* > 0.05). For these reasons, it is difficult to determine the potential DDI between DPZ and TAD at this time. Most PK-related DDIs occur through the induction or inhibition of metabolic enzymes, including CYP 450 and UDP-glucuronosyltransferase or drug transporters [43]. DPZ is primarily metabolized by CYP 2D6 and CYP 3A4, and then undergoes glucuronidation [44,45], whereas TAD is metabolized by CYP 2D9 and CYP 3A4. Although both drugs are metabolized by CYP3A4, PK-related DDI between DPZ and TAD has not been reported. However, more mechanistic studies and multiple dose administration for at least one week for pharmacokinetic or toxicokinetic studies should be conducted to clarify potential DDI secondary to the interference of the metabolic pathways of DPZ and TAD.

## 3. Materials and Methods

### 3.1. Materials

DPZ (purity 99.6%), TAD (purity 99.7%), and lansoprazole (LPZ; internal standard (IS), purity 99%) were purchased from Sigma-Aldrich (St. Louis, MO, USA). Water was prepared using an Aqua MAX Ultra 370 (YL Instrument, Gyeonggi-do, Republic of Korea) water purification system for all analyses. For HPLC analysis, methanol was purchased from Burdick & Jackson (Charlotte, NC, USA), and acetonitrile was obtained from Thermo Fisher Scientific (Waltham, MA, USA). Ammonium acetate and formic acid were used for the mobile phase buffer preparation. Ammonium acetate and formic acid were obtained from Sigma-Aldrich and from Wako Chemicals (Osaka, Japan), respectively.

### 3.2. Liquid Chromatographic Conditions

The LC-MS/MS procedure was adapted by utilizing an Exion LC AD system and a TQ 6500 mass spectrometer equipped with an electrospray ionization source (AB SCIEX, Framingham, MA, USA). An aliquot of 2 μL of all prepared samples was injected into the LC-MS/MS system, and chromatographic separation was conducted using a Kinetex C18 100 Å column (100 × 2.1 mm, 2.6 μm, Phenomenex, Torrance, CA, USA) at 40 °C. The pump flow rate was 0.25 mL/min, and the autosampler temperature was set at 4 °C. The mobile phase was composed of 2 mM ammonium acetate and 0.1% formic acid in acetonitrile with gradient elution. The predetermined gradient elution system was as follows: 0–1 min (35% B), 1–1.1 min (60% B), 1.1–2 min (60% B), 2–2.1 min (35% B), and 2.1–4 min (60% B).

### 3.3. Mass Spectrometric Conditions

The optimized MS/MS conditions for simultaneous quantification of DPZ, TAD, and LPZ (IS) were as follows: gas 1:40.0 psig, gas 2:30.0 psig, ion spray voltage: 5500.0 V, turbo heater temperature: 550.0 °C, entrance potential: 10.0 V, collision activation dissociation: 9 psig, curtain gas: 20 psig, cell exit potential: 10 V. The declustering potential and collision energy were applied at 60 and 35 V for DPZ, 100 and 20 V for TAD, and 50 and 20 V for LPZ, respectively. Data on DPZ, TAD, and LPZ were collected and quantified using the Analyst software version 1.6.3. (AB SCIEX, Framingham, MA, USA).

### 3.4. Preparation of Calibration Standards and Plasma Samples

Standard stock solutions of DPZ, TAD, and LPZ (IS) were prepared in methanol at a concentration of 100 µg/mL and stored at −20 °C. Working solutions were prepared by diluting the stock solutions with 50% methanol. The concentration of DPZ and TAD ranged from 1 to 200 ng/mL. Quality control (QC) samples were prepared at four different concentrations (1, 2, 16, and 160 ng/mL). The IS working solution was prepared by diluting the stock solution with 50% methanol to a concentration of 20 ng/mL. All working stock solutions were prepared and stored at −20 °C until use. Calibration standards and QC samples of DPZ and TAD in rat plasma were diluted with the corresponding working solutions and blank rat plasma. Drug-free plasma, that is, control (blank) plasma, stored in a deep freezer, was completely thawed at room temperature before use. Ten percent spiking with working DPZ and TAD stock solutions was performed in blank plasma. The spiked QC samples were stored at −70 °C for stability studies.

### 3.5. Preparation of Plasma Samples

The rat plasma was stored at −70 °C. The plasma samples were thawed at room temperature and then centrifuged at 1800× *g* (4 °C) for 8 min before pipetting. Using a micropipette, 30 µL of the plasma sample was transferred into a polypropylene microtube, and 10 µL of IS working solution was added to each sample. Further, 0.2 mL of acetonitrile was then added and vortexed for 1 min. The tube was then centrifuged at 22,285× *g* (4 °C) for 5 min. The supernatant was transferred to an analytical vessel, and 2 µL of the sample was injected into the LC-MS/MS system.

### 3.6. Protocol Validation

The established methods were validated by the required bioanalytical method validation guidelines published by the Ministry of Food and Drug Safety in Korea (MFDS) and the U.S. Food and Drug Administration (FDA) [26,27]. The established protocol was validated for selectivity, linearity, accuracy, precision, recovery, matrix effect, and stability in compliance with the bioanalytical method guidelines.

#### 3.6.1. Specificity and Lower Limit of Quantitation (LLOQ)

Selectivity was examined by analyzing six randomly selected blank rat plasma samples to investigate the presence of endogenous materials. These samples were prepared using the pretreatment procedure described in Section 3.5, and both the analytes and IS were free of interference with each other. LLOQ implies the lowest concentration of the analyte in rat plasma. The signal-to-noise (S/N) ratio was greater than 10, and the deviation from accuracy and precision was less than 20%.

#### 3.6.2. Linearity

Linearity was determined using the linear regression model (1/*x*^2^) for the calibration curve range of DPZ and TAD. The calibration curves were expressed as *y = ax + b*, where y represents the mean of the peak area ratios of the analytes to their internal standards, *a* represents the slope of the calibration curves, *b* represents the *y*-intercept of the calibration curve, and *x* represents the analyte concentration. Satisfactory linearity was determined using a correlation coefficient of 0.99 or greater.

#### 3.6.3. Precision and Accuracy

The precision and accuracy of the simulated quantification method were determined by concentration analysis of five replicates of each sample from the validation batch. For intra- and inter-day testing, four different concentrations of QC samples were prepared, and five sets of repeated sets were analyzed to determine the accuracy and precision for three consecutive days. The accuracy and precision of within- and between-run analyses should be within 15% of the nominal values, except for the LLOQ, which should be within 20% of the nominal value.

#### 3.6.4. Stability

Stability analyses of the stock solutions were performed at room temperature for 3 h and under refrigerated conditions (4 °C) for 14 days. The stability of the plasma analytes was tested for low and high QC under pre-specified conditions: room temperature for 5 h, refrigeration (4 °C) for 14 days, autosampler (10 °C) for 20 h, four cycles of freezing-thawing, and finally frozen conditions (−70 °C) for 28 days for long-term stability.

#### 3.6.5. Recovery and Matrix Effect

Recovery and matrix effects were measured by evaluating matrix-induced ion enhancement or suppression. The recovery was examined at three QC concentrations by comparing the mean analytical peak area of six replicates of each QC sample and the corresponding areas of the equivalent concentration of the analytes spiked into blank rat plasma after extraction. The matrix effect was determined by comparing the peak area of standard QC concentrations with the corresponding areas of the equivalent concentrations of the spiked analytes extracted from the blank rat plasma.

### 3.7. Application to a Pharmacokinetic Study

The pharmacokinetic (PK) tests in rats were approved by the Animal Ethics Committee (PK Protocol number: 21-KE-155; date of approval: March 2, 2021). Male Sprague-Dawley rats (5 weeks old) from KNOTUS Co., Ltd. (Incheon, Republic of Korea) were maintained under consistent laboratory conditions (temperature, 23 ± 3 °C; relative humidity, 55 ± 15%; light/dark cycle, 12 h). Rats were divided into three groups (*n* = 3): group 1, DPZ (3 mg/kg); group 2, TAD (0.3 mg/kg); and group 3, DPZ (3 mg/kg) combined with TAD (0.3 mg/kg). Animals were fasted overnight and allowed free access to water. DPZ powder and TAD powder in aqueous solutions were orally administered to rats at an equivalent DPZ dose of 2.26 mg/kg and TAD dose of 2.25 mg/kg (3 and 0.3 mg daily dosage converted to volume ratio, respectively. Blood samples (300 μL) were directly collected from the jugular vein using heparinized syringes at 0.5, 1, 2, 4, 6, 8, and 24 h. The collected blood samples were immediately centrifuged at 19,314× *g* for 2 min to prepare plasma samples, which were then stored at −70 °C until analysis.

### 3.8. Statistical Analysis of the Pharmacokinetic Study

The peak plasma concentration, *C*_max_, and *T*_max_ (the time to reach *C*_max_) for DPZ and TAD were calculated using individual plasma concentration–time profiles. A non-compartmental method for extravascular input provided in BA Cal 2007 software (MFDS, ver. 1.0.0) was used to calculate PK parameters, including AUC_last_ (area under the plasma drug concentration–time curve between 0 and 24 h) and extrapolated AUC_inf_ (AUC from 0 to infinity). The terminal half-life (*T*_1/2_) was calculated as ln 2/λ_z_ (λz is the elimination rate constant estimated by using the least-squares linear regression method of the terminal log-linear portion of the plasma concentration–time curve). The apparent clearance (CL/F) was calculated as CL/F = dose/AUC_0-∞._ The Mann–Whitney U test was used to determine the differences in the mean plasma concentration and PK profiles of DPZ and TAD between the single and combined administration groups. Statistical analysis was performed using SPSS Statistics (version 26.0; IBM SPSS Statistics for Windows, Armonk, NY, USA), and a *p*-value < 0.05 was considered statistically significant.

## 4. Conclusions

We established the first sensitive and simple LC-MS/MS-based analytical method for simultaneously quantifying DPZ and TAD in rat plasma. This LC-MS/MS analytical method was fully validated according to the guidelines published by the MFDS in Korea and the U.S. FDA and was successfully implemented in a PK study designed for DDI detection between DPZ and TAD in anticipation of their potential concurrent clinical use for AD management. Our analytical method for simultaneous quantification of DPZ and TAD achieved excellent performance and offered the following advantages over other previously reported single analyte quantification methods of either DPZ or TAD: the analysis was performed using a small plasma volume (30 µL) and a simple and rapid protein precipitation protocol, that is, high-throughput [27,35,36,37,46,47,48,49,50]. This study provided the first preliminary evidence of non-significant DDI between DPZ and TAD based on PK profiles in rats. The findings of this study may contribute to the current body of literature, especially for therapeutic drug monitoring associated with DDI; however, further detailed studies on the PK parameters of DPZ and TAD are still warranted to determine risks versus benefits, if any, associated with their concurrent administration.

## Figures and Tables

**Figure 1 molecules-28-02352-f001:**
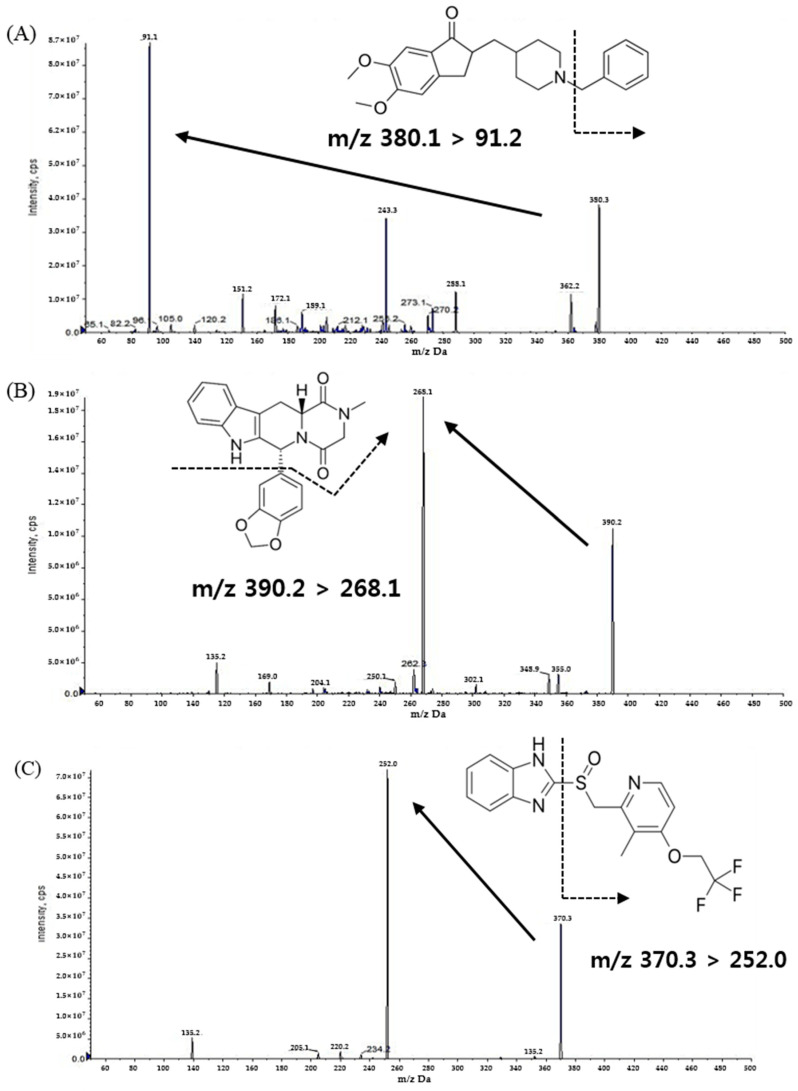
Product ion mass spectra and the pattern of fragmentation of (**A**) donepezil, (**B**) tadalafil, and (**C**) lansoprazole.

**Figure 2 molecules-28-02352-f002:**
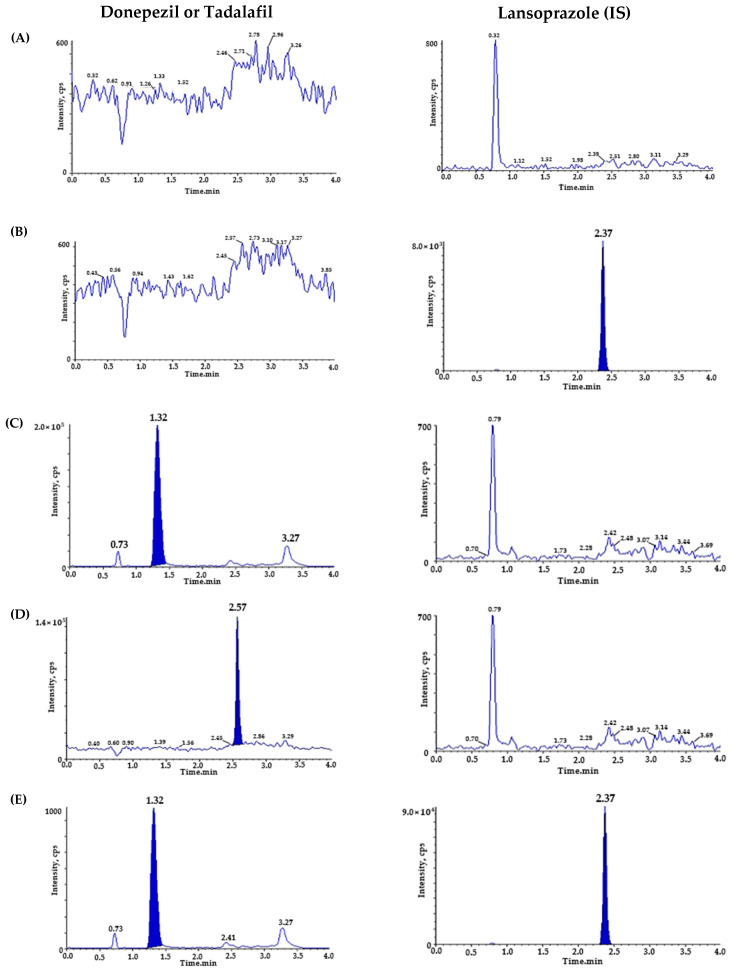
Representative chromatograms of (**A**) double blank plasma (without donepezil, tadalafil, and IS); (**B**) blank plasma spiked with lansoprazole (IS, 20 ng/mL); (**C**) blank plasma spiked with donepezil (ULOQ, 200 ng/mL); (**D**) blank plasma spiked with tadalafil (ULOQ, 200 ng/mL); (**E**) donepezil chromatogram of blank plasma spiked with donepezil, tadalafil (LLOQ, 1 ng/mL), and lansoprazole (IS, 20 ng/mL); (**F**) tadalafil chromatogram of blank plasma spiked with donepezil, tadalafil (LLOQ, 1 ng/mL), and lansoprazole (IS, 20 ng/mL); (**G**) donepezil chromatogram of sample plasma 1 h after oral administration of donepezil (3 mg/kg), measured concentration of donepezil (25.297 ng/mL) and tadalafil (0.3 mg/kg); and (**H**) tadalafil chromatogram of sample plasma 1 h after oral administration of donepezil (3 mg/kg) and tadalafil (0.3 mg/kg, measured tadalafil concentration: 12.943 ng/mL).

**Figure 3 molecules-28-02352-f003:**
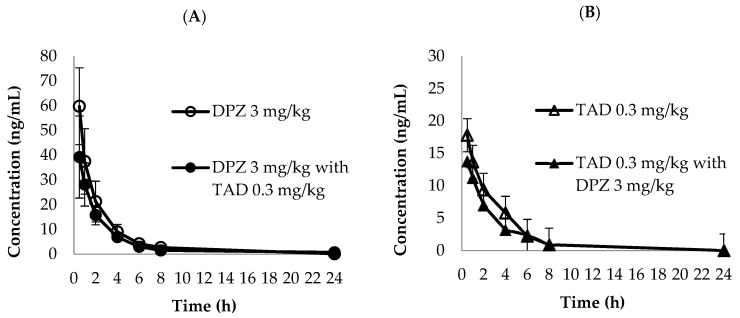
Mean plasma concentration–time curves of (**A**) donepezil (3 mg/kg, ○: single dose, ●: combined with tadalafil) and (**B**) tadalafil (0.3 mg/kg, △: single dose, ▲: combined with donepezil) in PK tests (*n* = 3).

**Table 1 molecules-28-02352-t001:** Intra- and inter-day precision and accuracy of the LC-MS/MS method for determining donepezil and tadalafil concentrations in rat plasma.

Compound	Nominal Concentration (ng/mL)	Intra-Day (*n* = 5)			Inter-Day (*n* = 5)		
		Mean ± SD (ng/mL)	Precision (CV, %) ^a^	Accuracy (%) ^b^	Mean ± SD (ng/mL)	Precision (CV, %)	Accuracy (%)
Donepezil	1	0.93 ± 0.02	2.15	93.10	0.94 ± 0.04	3.84	93.80
	2	1.99 ± 0.02	1.06	99.30	2.01 ± 0.07	3.33	100.65
	16	16.35 ± 0.25	1.55	102.17	16.40 ± 0.52	3.14	102.53
	160	139.08 ± 3.15	2.26	86.93	142.53 ± 6.77	4.75	89.08
Tadalafil	1	0.96 ± 0.05	5.52	96.10	0.97 ± 0.08	8.19	96.50
	2	1.92 ± 0.06	3.18	96.05	1.94 ± 0.11	5.51	97.05
	16	15.95 ± 0.50	3.10	99.66	15.92 ± 0.54	3.42	99.48
	160	153.29 ± 2.91	1.90	95.80	155.40 ± 10.44	6.72	97.12

^a^ CV (%) = (standard deviation of the calculated concentrations/mean concentration) × 100. ^b^ Accuracy (%) = (predicted concentration/nominal concentration) × 100.

**Table 2 molecules-28-02352-t002:** Stability of donepezil and tadalafil stock solutions in rat plasma under different conditions.

Stability Storage Condition	Donepezil		Tadalafil	
	Nominal Concentration (ng/mL)	(Mean ± SD, %)	Nominal Concentration (ng/mL)	(Mean ± SD, %)
Stock Solution Stability				
Room temperature (3 h)		103.43 ± 0.58		107.75 ± 10.32
Refrigeration (4 °C, 14 days)		103.53 ± 1.71		106.30 ± 5.99
Plasma Sample Stability				
Room temperature (5 h)	2	98.05 ± 4.42	2	95.75 ± 4.54
	160	89.84 ± 1.95	160	95.18 ± 0.41
Refrigeration (4 °C, 14 days)	2	93.33 ± 4.72	2	93.52 ± 3.03
	160	97.29 ± 2.34	160	107.44 ± 5.73
Freeze-thaw stability (4 cycles)	2	98.05 ± 0.40	2	95.15 ± 2.69
	160	92.54 ± 1.34	160	103.60 ± 1.55
Autosampler (10 °C, 20 h)	2	98.60 ± 2.46	2	99.63 ± 6.03
	160	98.19 ± 4.87	160	102.18 ± 3.98
−70 °C (28 days)	2	107.57 ± 3.61	2	99.17 ± 3.88
	160	93.41 ± 3.90	160	104.42 ± 4.78

**Table 3 molecules-28-02352-t003:** Extraction recovery and matrix factor for donepezil, tadalafil, and lansoprazole (IS) from rat plasma.

Compound	Nominal Concentration (ng/mL)	Recovery (%) ^a^		Matrix Effect (%) ^b^	
		Mean ± SD (%)	Precision (CV, %)	Mean ± SD (%)	Precision (CV, %)
Donepezil	2	95.11 ± 2.59	2.72	97.05 ± 1.67	1.72
	16	90.86 ± 4.33	4.77		
	160	95.99 ± 1.80	1.88	108.78 ± 2.87	2.64
Tadalafil	2	97.74 ± 11.05	11.31	58.46 ± 3.58	6.13
	16	96.17 ± 5.34	5.55		
	160	98.59 ± 3.93	3.99	56.16 ± 6.48	11.55
IS	20	97.96 ± 3.40	3.47	90.90 ± 4.55	5.01

Data are presented as mean ± SD (*n* = 6). ^a^ Extraction recovery (%) = [(peak area of analyte spiked before extraction)/(peak area of analyte spiked after extraction)] × 100. ^b^ Matrix effect (%) = [(peak area of analyte spiked after extraction)/(peak area of analyte in the pure standard solution)] × 100.

**Table 4 molecules-28-02352-t004:** Pharmacokinetic parameters of donepezil and tadalafil in rat plasma after oral administration of a single dose and combined doses (*n* = 3).

Parameter	Donepezil (3 mg/kg)		Tadalafil (0.3 mg/kg)	
	Alone	with TAD	Alone	with DPZ
	Mean ± SD	Mean ± SD	Mean ± SD	Mean ± SD
*C*_max_ (ng/mL)	59.776 ± 15.502	39.235 ± 16.562	17.795 ± 5.998	13.750 ± 9.160
AUC_last_ (h·ng/mL)	129.583 ± 38.090	100.500 ± 29.467	46.987 ± 24.384	34.419 ± 27.372
AUC_inf_ (h·ng/mL)	133.901 ± 37.878	106.379 ± 32.097	50.362 ± 23.612	38.931 ± 26.821
*T*_max_ (h)	0.500 ± 0.000	0.500 ± 0.000	0.500 ± 0.000	0.500 ± 0.000
*CL*/F (L/h)	2.382 ± 0.759	2.980 ± 0.888	7.355 ± 4.516	11.819 ± 9.852
*T*_1/2_ (h)	2.630 ± 1.302	3.793 ± 2.032	1.517 ± 0.120	1.962 ± 0.212
*MRT*_last_ (h)	2.617 ± 0.923	3.735 ± 1.602	2.532 ± 0.214	3.079 ± 0.444

*C*_max_, peak plasma concentration; AUC_last_, area under the plasma drug concentration–time curve to the last measurable time; AUC_inf_, area under the plasma drug concentration–time curve to infinity; *T*_max_, time to reach peak plasma concentration; *CL*_(inf)_/F, apparent systemic clearance; *T*_1/2,_ terminal elimination half-life; *MRT*_last_, mean residence time from the first observed to last measurable concentration.

## Data Availability

The data presented in this study are available upon request.

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
