# Peer review of "Development of a Rapid LC-MS/MS Method for Simultaneous Quantification of Donepezil and Tadalafil in Rat Plasma: Its Application in a Pharmacokinetic Interaction Study after Oral Administration in Rats"

_molecules, 2023, doi:10.3390/molecules28052352_

Round 1

Reviewer 1 Report

This manuscript described the novel assay for the determination of simultaneous Quantification of Donepezil and Tadalafil in Rat Plasma by UPLC-MS/MS method. Developed assay has been validated and was applied in pharmacokinetic study in rats. Authors claim novelty as this is first assay for simultaneous quantification of both analytes in biologicals. Overall, the manuscript is written well and organised but it require appropriate revision before consideration.

1.    Title of the manuscript required to change as application has been performed in intention to interaction finding. So it require to add as application in PK interaction studies.

2.    Why the author selected tadalafil for this study? There are many other drug also available as PDE5 inhibitor.

3.    A significant amount of ion suppression effects  (matrix-effects) has been observed (56-58%) for tadalafil? But authors confirmed that the matrix effects are negligible for this method. Kindly justify it.

4.    The result confirmed that there is no significant interaction between these drug co-administration. But I think the designed of this study was not appropriate. How we can expect the enzyme induction/inhibition by single dose administration? Authors need to designed 4 group studies two for single dose each analyte and two for multiple dose for atleast one week for each drug followed by single dose administration. Also n=? not mentioned in this study.

5.    Authors should not evaluated other parameters of PK e.g. half-life, MRT, Clearence etc.

6.    In sample preparation 30 uL of sample was diluted with 210 ul of acetonitrile. And sample was injected to lc-ms/ms directly without sample drying. Then how the sencitivity upto 1ng/mL was achieved after 7 times dilution?

7.    How the dose of tadalafil and donepezil was decided in this study.

Author Response

February 24, 2023

We would like to thank the reviewers for their thoughtful and constructive comments regarding our manuscript. Below, we address point by point how we respond to the reviewer’s comments which are reflected with the highlight in the revised manuscript, and we hope that the revised manuscript is now adequate for publication in Molecules.

Reviewer’s comments:

Reviewer 1:

  1. Title of the manuscript required to change as application has been performed in intention to interaction finding. So it require to add as application in PK interaction studies.

Response: To follow your advice, we changed the title below in the revised manuscript.

“Development of a Rapid LC-MS/MS Method for Simultaneous Quantification of Donepezil and Tadalafil in Rat Plasma: its Application in Pharmacokinetic Interaction Studies after Oral Administration in Rats”.

  1. Why the author selected tadalafil for this study? There are many other drugs also available as PDE5 inhibitor.

Response: Tadalafil was chosen for these reasons. (i) It was found that donepezil combined with tadalafil (even with least drug use) in animal PD models showed strong synergistic pharmacodynamic effects to improve Alzheimer's disease or cognitive function in animal models (https://patentimages.storage.googleapis.com /f2/a1/ d3/8f0cb034ff2e5b /KR102393649B1.pdf). (ii) Although PDE inhibitors such as sildenafil (Viagra), tadalafil (Cialis), vardenafil (Levitra),  and avanafil (Stendra) are major drugs used in the clinical trials, we chosed tadalafil based on the amount of drug used and duration of action.

  1. A significant amount of ion suppression effects (matrix-effects) has been observed (56-58%) for tadalafil? But authors confirmed that the matrix effects are negligible for this method. Kindly justify it.

Response: Thanks for your comment. We absolutely agreed with your comment and deleted the word “negligible” in the revised manuscript. As the protein precipitation method for simultaneous determination of donepezil and tadalafil is used, a significant matrix effect can be expected compared to the SPE and LLE methods. Although the matrix effect was significant, it showed high reproducibility with the high signal intensity and reproducibility of the TAD peak.

  1. The result confirmed that there is no significant interaction between these drug co-administration. But I think the designed of this study was not appropriate. How we can expect the enzyme induction/inhibition by single dose administration? Authors need to designed 4 group studies two for single dose each analyte and two for multiple dose for at least one week for each drug followed by single dose administration. Also n=? not mentioned in this study.

Response: Thanks for your advice. In the paper, we mentioned that rats were divided into three groups (n=3) and described this in Figure 3 and Table 4. We wanted to focus on showing the first development for the simultaneous quantification LC-MS/MS methods of donepezil and tadalafil and then applied this to pharmacokinetic DDI studies for the druggable feasibility. Therefore, based on your advice, we will explain in the results and discussion section that ‘more mechanistic studies and multiple doses administration for at least one week for pharmacokinetic or toxicokinetic s study should be conducted to clarify potential DDI secondary to the interference of the metabolic pathways of DPZ and TAD’.

  1. Authors should not evaluated other parameters of PK e.g. half-life, MRT, Clearence etc.

Response: Thanks for your advice. We added the PK parameters of half-life, MRT, and clearence in the revised manuscript.

  1. In sample preparation 30 uL of sample was diluted with 210 ul of acetonitrile. And sample was injected to lc-ms/ms directly without sample drying. Then how the sencitivity upto 1ng/mL was achieved after 7 times dilution?

Response: Our developed assay method is to determine the concentration by injecting 2 µL after deproteinization by adding 200 µL of acetonitrile to 30 µL of plasma. The calibration curve preprocessing method and the sample preprocessing method for determining the drug concentration were applied in the same process. In addition, the instrument used is a TQ 6500 mass spectrometer, which has excellent sensitivity, so even if the injection volume is reduced to 2 µL, a 1 ng/mL measurement is possible.

  1. How the dose of tadalafil and donepezil was decided in this study

Response: Based on the screening data for Alzheimer's disease or cognitive dysfunction using various combinations of two drugs, we found that 3 mg/ of donepezil and 0.3 mg tadalafil (ratio 10:1) showed the good efficacy in the mice (https://patentimages.storage.googleapis.com/f2/a1/d3/8f0cb034ff2e5b /KR102393649B1.pdf).

Thank you very much for your good advice.

Sincerely yours,

Kyung-Tae Lee, Ph.D.

Department of Pharmaceutical Biochemistry, College of Pharmacy

Kyung Hee University

26, Kyungheedae-ro, Dongdaemun-gu, Seoul, 02447, Seoul, Republic of Korea

References

  1. Neurorive Inc., Ltd.(2021), KR102393649B1. 189 Cheongsa-ro, Seo-gu, Daejeon: Korea Patent Office https://patentimages.storage.googleapis.com/f2/a1/d3/8f0cb034ff2e5b/KR102393649B1.pdf

Reviewer 2 Report

The paper is devoted to the development of the LC-MS/MS method for the Donepezil and Tadalafil quantification. Although the importance of such a method is unquestionable, there is little if any method development (e.g. LC or extraction optimization) described. The paper basically contains a proved statement 'LC-MS/MS can be used to this end'.

Major points:

Section 2.2.2. The data on linearity should be presented. In addition, based on the Section 2.2.3 data, the method substantially deviates from linearity already at the concentration of 160 ng/ml

The Discussion section is missing, and the limits, speed, and detected matrix effects are not discussed in the text

Minor points:

Figure 2: the left and right chromatogram on each pane should be either labeled, or explained in the figure caption. The numeric peaks labels meaning should be explained
Table 2. The data format seems a bit confusing. Maybe it is better to show both the detected concentration and its percent amount?
Table 3. The matrix effect for Tadalafil seems to be quite substantial, but it is stated as 'negligible'. This should be discussed in detail
Methods: Preparation of plasma samples. The parameters of centrifugation should be added
There are no data on the existing methods for the Donepezil and Tadalafil detection and quantification

Author Response

February 24, 2023

We would like to thank the reviewers for their thoughtful and constructive comments regarding our manuscript. Below, we address point by point how we respond to the reviewer’s comments which are reflected with the highlight in the revised manuscript, and we hope that the revised manuscript is now adequate for publication in Molecules.

Reviewer’s comments:

Reviewer 2:

The paper is devoted to the development of the LC-MS/MS method for the Donepezil and Tadalafil quantification. Although the importance of such a method is unquestionable, there is little if any method development (e.g. LC or extraction optimization) described. The paper basically contains a proved statement 'LC-MS/MS can be used to this end'.

Major points:

Section 2.2.2. The data on linearity should be presented.

In addition, based on the Section 2.2.3 data, the method substantially deviates from linearity already at the concentration of 160 ng/Ml

Response: (1) We added the equation for the calibration curves (n = 5) with the mean ± SD of the slope in the revised manuscript. (2) Concentrations of donepezil at 160 ng/mL of intra- and inter-day showed 139.08 ± 3.15 and 142.53 ± 6.77 ng/mL, respectively, and seemed substantially deviation from linearity. However, this high concentration does not exceed the precision (Intra-day, 2.26%; Interday, 4.75%) and accuracy (Intra-day, 86.93%; Interday, 89.08%) which satisfied the precision and accuracy ranges (within ± 15%), as specified in the guidance of the MFDS and the FDA for bioanalytical applications ranges (less than 15%) even though these concentrations seemed to be reaching saturation.

The Discussion section is missing, and the limits, speed, and detected matrix effects are not discussed in the text

Response: Thanks for your advice. We changed the title from results to results and discussion. We added the discussion contents with yellow highlights in the revised manuscript.

Major points:

Figure 2: the left and right chromatogram on each pane should be either labeled, or explained in the figure caption. The numeric peaks labels meaning should be explained

Response: To follow your advice, we add labels in Figure 2 of the revised manuscript.

Table 2. The data format seems a bit confusing. Maybe it is better to show both the detected concentration and its percent amount?

Response: Although the detected concentration is useful, it can be calculated based on the peak area response of the stored samples was directly compared with that of fresh stock solutions. Therefore nominal concentration and % are better to compare in the stability test based on our references (PMID: 32977631, PMID: 34834083, PMID: 30802201, PMID: 22771234)

Table 3. The matrix effect for Tadalafil seems to be quite substantial, but it is stated as 'negligible'. This should be discussed in detail

Response: Thanks for your comment. We absolutely agreed with your comment and deleted the word “negligible” in the revised manuscript. As the protein precipitation method for simultaneous determination of donepezil and tadalafil is used, a significant matrix effect can be expected compared to the SPE and LLE methods. Although the matrix effect was significant, it showed high reproducibility with the high signal intensity and reproducibility of the TAD peak.

Methods: Preparation of plasma samples. The parameters of centrifugation should be added

Response: We added the parameters of centrifugation “at 1,800 × g (4 °C) for 8 min” in the revised manuscript.

There are no data on the existing methods for the Donepezil and Tadalafil detection and quantification

explain the data of existing methods for Donepezil and Tadalafil detection and quantification in discussion section.

Response:  We added the existing methods for Donepezil and Tadalafil detection and quantification in the introduction part and conclusion part of the revised manuscript.

Thank you very much for your good advice.

Sincerely yours,

Kyung-Tae Lee, Ph.D.

Department of Pharmaceutical Biochemistry, College of Pharmacy

Kyung Hee University

26, Kyungheedae-ro, Dongdaemun-gu, Seoul, 02447, Seoul, Republic of Korea

Round 2

Reviewer 1 Report

Authros addressed the comments properly and can be consider for publication

Reviewer 2 Report

The points were addressed